# High Prevalence of Scoliosis in a Large Cohort of Patients with Prader-Willi Syndrome

**DOI:** 10.3390/jcm11061574

**Published:** 2022-03-13

**Authors:** Antonino Crinò, Michela Armando, Marco Crostelli, Osvaldo Mazza, Dario Bruzzese, Alessio Convertino, Danilo Fintini, Sarah Bocchini, Sara Ciccone, Alessandro Sartorio, Graziano Grugni

**Affiliations:** 1Reference Center for Prader-Willi Syndrome, Bambino Gesù Children’s Hospital, Research Institute, Palidoro, 00050 Rome, Italy; convertinoalessio@gmail.com (A.C.); danilo.fintini@opbg.net (D.F.); sarah.bocchini@opbg.net (S.B.); sara.ciccone@auslromagna.it (S.C.); 2Paediatric Neurorehabilitation Unit, Bambino Gesù Children’s Hospital, Research Institute, Palidoro, 00050 Rome, Italy; armando@opbg.net; 3Spine Surgery Unit, Bambino Gesù Children’s Hospital, Research Institute, 00165 Rome, Italy; marco.crostelli@opbg.net (M.C.); osvaldo.mazza@opbg.net (O.M.); 4Department of Public Health, University of Naples “Federico II”, 80131 Naples, Italy; dbruzzes@unina.it; 5Division of Auxology, Istituto Auxologico Italiano, IRCCS, 28824 Piancavallo, Italy; sartorio@auxologico.it (A.S.); g.grugni@auxologico.it (G.G.); 6Experimental Laboratory for Auxo-Endocrinological Research, Istituto Auxologico Italiano, IRCCS, 28824 Piancavallo, Italy

**Keywords:** Prader–Willi syndrome, scoliosis, obesity, growth hormone therapy

## Abstract

The characteristics of scoliosis were investigated in a large cohort of children and adults with Prader–Willi syndrome (PWS), analysing the role of age, gender, puberty, body mass index (BMI), genotype and growth hormone therapy (GHT) on its onset and severity. A retrospective cross-sectional study was performed in 180 patients with genetically confirmed PWS (96 females), aged 17.6 ± 12 years. Eighty-five subjects (47%) were obese. One hundred and fifty subjects (83.3%) were on GHT, while 30 patients had never been treated. Overall, 150 subjects (83.3%) were affected by scoliosis, 80.2% of children and adolescents and 87.8% of adults. A mild degree of scoliosis was observed in 58 patients (38.7%), moderate in 43 (28.7%) and severe in 49 (32.6%). Median age at diagnosis of scoliosis was 6.3 years, while the severe forms were diagnosed earlier (median age: 3.8 years). The cumulative probability at 5 years of age was equal to 0.403 and almost doubled at 15 years. No significant associations were found between scoliosis and genotype, gender, pubertal stage, GHT and BMI. A corset was prescribed to 75 subjects (50%) at a median age of 7.5 years, while 26 subjects (17.3%) underwent surgery at a median age of 13.1 years. Our data indicate that scoliosis is one of the major concerns for PWS patients that increases with age, and therefore suggest the need for regular systematic monitoring of spinal deformity from paediatric age.

## 1. Introduction

Prader–Willi syndrome (PWS) is a rare genetic disorder, with an incidence of approximately 1 in 21,000 newborns [1]. It is considered the most common syndromic cause of morbid obesity and is caused by abnormalities of chromosome 15, including paternal deletion (del15), uniparental maternal disomy (UPD15) and, more rarely, imprinting centre defect [2].

A complex hypothalamic-pituitary dysregulation is currently thought to be partly responsible for PWS phenotype. The predominant features include neonatal hypotonia, poor feeding and lack of appetite in infancy, followed by hyperphagia with early childhood-onset morbid obesity (if uncontrolled), dysmorphic features (characteristic facial appearance, small hands and feet), behavioural problems, cognitive impairment, multiple endocrine abnormalities (hypogonadism, growth hormone/insulin-like growth factor-I axis dysfunction, hypothyroidism, hyperghrelinemia and, rarely, central adrenal insufficiency), and sleep-disordered breathing [3,4].

Musculoskeletal manifestations are commonly observed in PWS, including scoliosis, kyphosis, hip dysplasia, ligamentous laxity, osteoporosis and various lower limb anomalies [5]. Incomplete pubertal development and low sex hormone levels are believed to be major causes of osteoporosis. Obesity, bone dysplasia and hypotonia of paravertebral muscles have been proposed as risk factors for the development of scoliosis [6,7]. In addition, a recent study has suggested that the elevated ghrelin levels may play a role in the pathophysiology of early-onset scoliosis in subjects with PWS [8]. The age of diagnosis of scoliosis is reported to follow a bimodal distribution pattern, with the first peak before the age of four and the second one in the adolescent period [9,10]. However, clinical diagnosis of scoliosis and other spinal deformities are often delayed in PWS children, probably due to obesity and/or hypotonia [10] and to the lack of vertebral rotation related to the curve [11], making it one of the major health problems in adulthood [12,13].

Despite these diagnostic difficulties, scoliosis has been reported frequently in PWS, although its prevalence is not well established to date, ranging from 15% to 86% of cases [11,14]. Similarly, the frequency of moderate/severe scoliosis is uncertain, due to the fact that the available data are scarce and not homogeneous, depending on the different criteria adopted, ranging from 9% to 50% of cases [10,15,16,17,18].

Age, gender and genotype are possible risk factors for the onset and progression of scoliosis in PWS. Typically, scoliosis increases with age, ranging from occurring in 23% of PWS children before 4 years to 74% in adult subjects [10,13]. As far as gender is concerned, both an increased likelihood of developing scoliosis in females [11,19] and a lack of gender differences were reported [6,10]. Comparing the prevalence of scoliosis between the two major genotypic subtypes, a higher risk of developing scoliosis was found both in patients with UPD15 [11] and in those with del15 [6,13], while other reports failed to detect differences between the genotypic subtypes [10,20]. These discrepancies, together with the discordant results about severity of scoliosis, might be related to the different clinical characteristics of the study groups, including sample size, age, and degree of obesity, as well as duration of follow-up.

Since scoliosis is characterised by a rapid evolution with growth [21,22], GH therapy (GHT) might be advocated as a possible risk factor for its onset and progression. However, both short-term and long-term analysis demonstrated the lack of adverse effects of GHT on the prevalence and severity of scoliosis in children with PWS [17,23].

With this background, the purpose of the present study was to investigate the characteristics of scoliosis in a large cohort of children and adults with PWS, analysing the role of age, gender, puberty, body mass index (BMI), genotype and GHT on its onset and severity.

## 2. Patients and Methods

### 2.1. Study Population

This is a retrospective, cross-sectional study conducted from June 2018 to September 2020 at Bambino Gesù Children’s Hospital, Palidoro (Rome), Italy, and Istituto Auxologico Italiano, Verbania, Italy. The study population included 180 patients (96 females and 84 males, aged 1.7–49.7 years) with PWS. The inclusion criteria were the availability of all genetic, auxological and instrumental data. For each subject, medical and surgical data were retrospectively analysed from medical records, including age at diagnosis of scoliosis and its management and evolution, genotype, age of onset of GHT and its duration. The patients were divided into two groups according to age at final evaluation: (1) children and adolescents (n. 106, <18 years) and (2) adults (n. 74, ≥18 years).

Both the Ethical Committee of the Bambino Gesù Children’s Hospital and the Ethical Committee of Istituto Auxologico Italiano approved the study protocol (research code: 2092/2020, acronym: PWS_ISS, and research code: 2014_01_21_06; acronym: SCOLADUPWS, respectively). Written informed consent was obtained from all parents or legal guardians, and from the patients when applicable. The study was performed in accordance with the Declaration of Helsinki (1975) and with the 2005 Additional Protocol to the European Convention of Human Rights and Medicine concerning Biomedical Research.

### 2.2. Auxological Data

All patients underwent body measurements wearing light underwear. Height was determined by a Harpenden Stadiometer (Holtain Limited, Crymych, Dyfed, UK). Body weight was measured to the nearest 0.1 kg, by using standard equipment. BMI was calculated as weight in kilograms divided by the square of the height in metres and expressed as standard deviation scores (SDS) [24] in order to normalise the values for age and sex. The cut-off value of BMI >2 SDS was used to define obesity in children and adolescents and 1.4–2 SDS for overweight. BMI from 18.5–25 kg/m^2^ was defined as normal, between 25–30 kg/m^2^ as overweight and above 30 kg/m^2^ confirmed obesity in adult patients, according to the World Health Organization criteria [25]. Pubertal development was assessed according to Tanner classification [26].

### 2.3. Scoliosis Assessment

All patients underwent observation of the standing and sitting posture, Adam’s forward bend test, and X-ray examination of the vertebral column (standing antero-posterior and lateral spinal radiographs). The Adam’s forward bend test is the standard clinical screening procedure for scoliosis, performed with the patients standing with their back to the observer and feet together, bending forward with knees flexed and arms extended (hanging).

Scoliosis assessment, including Cobb angle (CA) measurement was reviewed retrospectively by the same senior spine surgeon at Bambino Gesù Children’s Hospital, Rome. The CA is the angle between the two steepest vertebrae, i.e., the upper border of the upper vertebra in the curve and the lower border of the lower vertebra. Scoliosis was defined as a spinal curvature with a CA > 10° on a standing postero-anterior radiograph with vertical rotation and was classified using the Scoliosis Research Society classification [27]: mild (CA 10–20°), moderate (CA 20–40°) and severe (CA > 40°). Finally, a progressive curve is defined as an increase in CA > 5° within 6 months.

### 2.4. Statistical Analysis

Demographical and clinical characteristics were summarised using standard descriptive statistics: mean ± standard deviation in the case of numerical variables with a symmetric distribution and median with interquartile range in the case of variables showing a substantial skewness; absolute frequencies and percentage (%) were used in the case of categorical factors. Accordingly, comparisons between groups were based on the Student *T* test, the Mann–Whitney *U* test or the chi-square test, as appropriate. Differences among groups defined by the severity of the scoliosis (mild, moderate and severe) were assessed using parametric and non-parametric (Kruskal–Wallis test) ANOVA.

Median age at diagnosis of scoliosis was estimated using the Kaplan–Meier method and Cox regression was used to quantify the impact of GHT, genotype and gender on the risk of scoliosis. GHT was considered as a time invariant variable and patients who started GHT after the diagnosis of scoliosis were considered to be in the non-treated group. *p* values < 0.05 were considered statistically significant. All statistical analyses were performed using R platform (version 4.0.2) [28].

## 3. Results

### 3.1. Characteristics of the Cohort

The demographical, genetic and clinical characteristics of the entire population of the study, stratified by age, are shown in Table 1.

Genetic diagnosis of PWS was obtained at a median age of 0.8 years (range: 0.1 to 20.5 years). Ninety-nine subjects had del15, while 78 presented UPD15. A positive methylation test was found in three individuals, but the underlying genetic defect was not identified.

At the time of the cross-sectional analysis, the mean age was 17.6 ± 12 years (range 1.7 to 49.7), with the majority of patients being children and adolescents (58.9%) and females (53%). Sixty-one patients were prepubertal (stage 1), 119 pubertal (stage 2–4) and none fully developed (stage 5). Altogether, 53 patients (29.4%) underwent treatment with sex steroids. Mean BMI was 29 ± 11.2 and mean BMI SDS 1.65 ± 1.59. According to BMI cut-offs, 85 subjects (47.2%) were considered obese (74.3% in adulthood and 28.3% in patients < 18 years).

One hundred and fifty patients (83%) had started GHT at a median age of 2.5 years, with a median duration of therapy of 6.3 years. Thirty subjects had never been treated with GH (Table 1).

### 3.2. Scoliosis

In the overall cohort, 150 patients (83.3%) developed scoliosis during a median follow-up of 33.5 years (min: 0.7 years; max 43.4 years). Median age at diagnosis of scoliosis was 6.3 years (95% C.I.: 5.6 to 7.9). Kaplan–Meier estimate of the cumulative probability of diagnosis of scoliosis in the overall cohort is shown in Figure 1, where the steepest slope is evident in the first 15 years of life. At 5 years of age, the cumulative probability was equal to 0.403 (95% CI: 0.325 to 0.472) and almost doubled (0.806; 95% CI 0.732 to 0.859) at the age of 15.

### 3.3. Predictive Factors for Scoliosis

Among the 150 patients who started GHT at any time of their follow-up, a subset of 111 patients (61.7%) received GHT before diagnosis of scoliosis at a median age of 1.8 years (range: 0.2 to 14.9 years), with a median duration of GHT of 6.7 years (range: 0.1 to 17.4 years). No significant difference was observed in this subset of patients with respect to the risk of developing scoliosis when compared with those who were never treated or those who started GHT after diagnosis of scoliosis. In particular, median age at scoliosis diagnosis was 6.25 years (95% CI: 5.3 to 8.3 years) in subjects undergoing GHT and 6.50 years (95% CI: 5.0 to 9.8 years) in untreated subjects with an HR of 1.08 (95% CI: 0.78 to 1.78; *p* = 0.655). Furthermore, neither genotype nor gender were significantly associated with the risk of developing scoliosis (Table 2).

### 3.4. Clinical and Anthropometric Measurements at Final Evaluation According to Scoliosis

At the time of the cross-sectional analysis, PWS subjects with and without scoliosis did not show any significant difference with respect to pubertal stages and BMI, while the patients with scoliosis were characterised by an older age (Table 3).

### 3.5. Severity of Scoliosis

At the time of the cross-sectional analysis, scoliosis was mild in 58 patients (38.7%), moderate in 43 subjects (28.7%) and severe in 49 individuals (32.6%). Associations between clinical and demographical factors and severity of scoliosis are reported in Table 4. A severe form of scoliosis was more prevalent in females (*p* = 0.025), while no significant difference was found with respect to genetics or to GHT. Severe forms of scoliosis were diagnosed at a younger age (3.8 years; range: 0.7 to 22.2) rather than moderate (6.5 years; range: 0.9 to 30.4; *p* = 0.006) and mild forms (6.2 year; range: 0.8 to 43.4; *p* = 0.005). Subjects with severe scoliosis did not show any significant difference with respect to BMI and BMI SDS when compared to patients with moderate or mild scoliosis. The prevalence of obesity was also similar in the three groups (*p* = 0.362) (Table 4).

### 3.6. Scoliosis Treatment

Among 150 patients with scoliosis, 75 patients (50%; 95% CI: 41.7% to 58.3%) required the use of a corset at a median age of 7.5 years (range: 1.1 to 15.2 years). Median duration of corset wearing was 5 years (range: 0.2 to 13.9 years).

Twenty-six patients (17.3%; 95% CI: 11.6% to 24.4%) underwent surgery (Table 5). Before undergoing surgery, in all patients, the CA was >45° (severe scoliosis) with a high risk of progression. Overall, scoliosis was diagnosed at a median age of 3.4 years (range 0.9 to 14.5 years). Median age at surgery was 13.1 years (range: 5 to 27.5 years). At the time of the cross-sectional analysis (median age: 16.2; range: 4.4 to 34.8), mean BMI and mean BMI SDS were 29.3 ± 8.4 and 1.75 ± 1.22, respectively. Twenty-four of these patients (92.3%) started GHT before surgery, at a median age of 2.7 years (range: 0.5 to 16 years) and for the duration of 9.3 ± 4.05 years (range: 1.3 to 16.8 years).

## 4. Discussion and Conclusions

Scoliosis was included as a supportive feature in the consensus clinical criteria of PWS, with no direct impact on the diagnostic score established by Holm et al. in 1993 [29,30]. This spinal deformity has long been underestimated in PWS, but with the improvement in early diagnosis and the multidisciplinary therapeutic approach, it has been diagnosed precociously and has become one of the major concerns for these patients. Scoliosis is considered to be a significant comorbidity for cardiopulmonary impairment, due to associated chest deformities [31,32]. The abnormal sideways curve of the spine is believed to be responsible for the high risk of respiratory complications associated with PWS, along with chest muscle weakness, muscle hypotonia in the upper airway, and obesity [33]. However, a severe and progressive scoliosis may represent a life-threatening deformity by itself [10,32]. Therefore, a prompt diagnosis of spinal deformities and recognition of associated factors are mandatory in these patients, in order to plan the most appropriate therapeutic and rehabilitative interventions.

The available literature on the prevalence of scoliosis in PWS patients has provided controversial results so far. Taking into account the studies with a significant number of patients, it was reported to occur in between 30% and 78% of cases. Nakamura et al. [34] noted that 58 out of 193 individuals (30%) had scoliosis. Likewise, an overall rate of 37.5%, which increased with age, was observed in a cohort of 96 children with PWS [10]. In a retrospective study performed in 145 children with PWS, Odent et al. [6] found an overall prevalence of scoliosis of 43%, which became 68% for those who had reached skeletal maturity. In a survey of 232 adults with PWS, nearly 50% showed orthopaedic problems (mostly scoliosis) [35]. This percentage was similar to that reported by Butler et al., who reviewed 14 previous studies that included patients of all age groups [36]. More recently, scoliosis was reported as the most prevalent somatic comorbidity (78%) in a cohort of 154 subjects with a median age of 27 years [37]. In line with these results, Pellikaan et al. showed a prevalence of scoliosis of 74% in a group of 115 young adults with PWS [13].

Our study adds further evidence that scoliosis affects the majority of patients with PWS. We found in our cohort of 180 subjects a high prevalence of scoliosis (83.3%), similar to the values observed in adult individuals [13,37]. This is not surprising, given that the majority of our patients are adolescents and adults. Thus, our data confirmed that the cumulative probability of scoliosis in PWS increases with age [6,10,19].

Apart from age, we found no association with gender, genotype and BMI or for pubertal stage. The lack of relationship with BMI seems to indicate that weight status is not a causative factor for scoliosis development. This finding has been previously reported both in children and in adults with PWS [6,15]. On the contrary, de Lind van Wijngaarden et al. found a significantly higher BMI SDS in children with scoliosis than in those without spinal deformity [10]. These differences might be related to the different degree of weight excess of the study groups. However, with the data available from the present study, we are actually unable to give definitive answers to this discrepancy.

Our study shows that the frequency of scoliosis was not statistically different at any age between patients who received GHT and those not treated. GHT is part of the current therapeutic regimen for patients with PWS [38]. Long-term GHT has been associated with improvement in height, dysmorphic features, muscle strength, body composition, psychomotor development and cognition in children with PWS [4]. In this context, there have been concerns in the past about the negative role of GHT on the onset or worsening of scoliosis. However, all studies evaluating both short- and long-term GHT, including our retrospective analysis, have shown a lack of adverse effects on spinal deformities [6,16,17,19,22,39]. Based on these findings, it is clear that scoliosis should not be considered a contraindication for initiation of GHT nor a reason to discontinue its administration or to reduce the dose in PWS children with scoliosis.

Our current results show a high prevalence of moderate and severe forms of scoliosis, which required timely conservative treatment or surgical approach even at a young age. Overall, 92/150 (61.3%) patients with scoliosis had a moderate/severe degree of spinal deformity, while 49 of them had CA > 40°. These values are greater than those observed in the literature [10,15,16,17,18]. However, it is not easy to compare our results with those of other reports. With the exception of Nakamura et al. [16], no other study has in fact clearly divided subjects with scoliosis according to the degree of severity, as established by the Scoliosis Research Society [27]. In spite of this, this finding is currently difficult to explain, and further longitudinal studies are needed to better understand this crucial point. In this context, our report is the first to document a gender difference in the prevalence of severe forms of scoliosis, being higher in females. This issue will require additional research. If confirmed, this observation could be useful in better defining a diagnostic–therapeutic approach tailored to these patients.

Among patients with moderate/severe scoliosis, spinal deformity deteriorated and required appropriate and specific treatment in a significant percentage of cases. Half of the subjects required timely conservative therapy, while for about one out of six of them the brace was unsuccessful and surgical treatment was required. It should be noted that the diagnosis of severe scoliosis in our cohort was made at a younger age than the moderate and mild forms. As a result, surgically treated subjects were also diagnosed with scoliosis at an early age. According to the literature, these data highlight the need to promptly detect scoliosis in the first months/years of age, through a timely and systematic clinical and instrumental follow-up [11,20]. In particular, a radiographic screening should be started as soon as the child begins sitting independently.

The strength of our study resides in the large sample size, taking into account that PWS is a rare disease, which also allowed subgroups to be stratified. In addition, the X-ray examination of the spine was carried out by the same senior spine surgeon, thus avoiding the risk of a significant intra-operator variability. However, there are also some limitations in our study. Firstly, one limitation is the retrospective cross-sectional design of the study, which does not allow us to draw conclusions about the natural history of scoliosis in PWS, particularly at the time of its appearance and its evolution during the life span of these individuals. A second weakness is the multicentre recruitment of our subjects, which makes the interpretation of the data less reliable than for those obtained in a single centre. However, enrolment and examination of patients were performed by a well-trained and experienced team from the two PWS centres at the third level.

Notwithstanding these limitations, our results could be important from a therapeutic and rehabilitative point of view, adding value in the clinical and therapeutic management of PWS and increasing awareness of this rare pathological condition.

In conclusion, our retrospective study confirms that scoliosis is one of the most frequent morbidities of PWS at any age. Furthermore, the well-established lack of adverse effects of GHT on the development and severity of scoliosis was also evident in our PWS cohort. For the first time, a high frequency of moderate and severe forms of scoliosis was demonstrated, the latter being more frequently detectable in the first years of life. This finding underlines the need for early radiological skeletal surveys once a child can sit unassisted along with a thorough clinical examination, as scoliosis may not be well recognised in the presence of obesity and/or hypotonia. Likewise, regular monitoring for development and/or progression of the spinal deformity is of utmost importance, given the frequent increase that occurs with the increase in age.

## Figures and Tables

**Figure 1 jcm-11-01574-f001:**
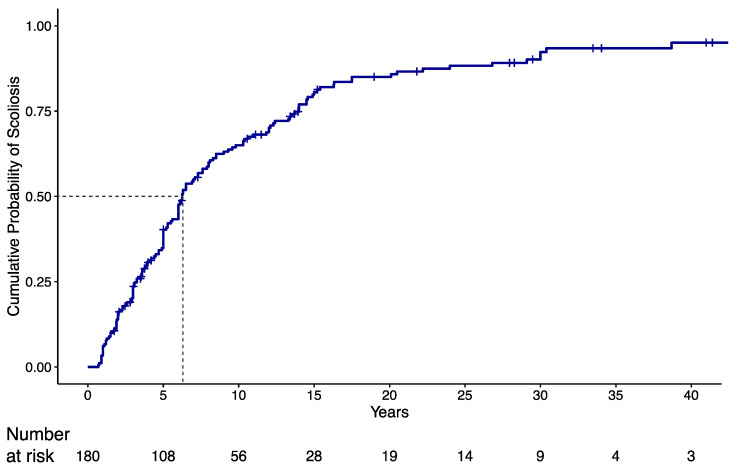
Kaplan–Meier (KMI) estimate of cumulative probability of scoliosis diagnosis in the overall cohort. Dotted line refers to the estimated median time to scoliosis diagnosis, i.e., the time interval for which the probability of getting scoliosis at the end of the interval is 0.5.

**Table 1 jcm-11-01574-t001:** Demographical, genetic and clinical characteristics of the entire population of the study stratified by age.

	Overall Cohort	Children (Age < 18 yrs)	Adults (Age ≥ 18 yrs)
**Number**	180	106 (58.9%)	74 (41.1%)
**Age at evaluation; yrs**	17.6 ± 12 (1.7 to 49.7)	9.3 ± 5 (1.7 to 17.9)	29.5 ± 8.6 (18 to 49.7)
**Age at PWS diagnosis; yrs**	0.8 [0.2; 3.6] (0.1 to 20.5)	0.2 [0.1; 0.9] (0.1 to 15)	3.2 [1; 10] (0.1 to 20.5)
**Genetics**			
*del15*	99 (55%)	53 (50%)	46 (62.2%)
*UPD15*	78 (43.3%)	50 (47.2%)	28 (37.8%)
*Other*	3 (1.7%)	3 (2.8%)	0 (0%)
**Gender**			
*Male*	84 (46.7%)	50 (47.2%)	34 (45.9%)
*Female*	96 (53.3%)	56 (52.8%)	40 (54.1%)
**Pubertal status** *(Tanner)*			
*1*	61 (33.9%)	61 (57.5%)	0 (0%)
*2*	27 (15%)	23 (21.7%)	4 (5.4%)
*3*	45 (25%)	16 (15.1%)	29 (39.2%)
*4*	47 (26.1%)	6 (5.7%)	41 (55.4%)
**BMI** (kg/m^2^)	29 ± 11.2 (12.5 to 64.6)	24.1 ± 9.6 (12.5 to 64.6)	36.1 ± 9.4 (21.7 to 59.7)
**BMI SDS**	1.65 ± 1.59 (−3.48 to 4.48)	1.04 ± 1.63 (−3.84 to 4.48)	2.53 ± 1.03 (−0.2 to 4.01)
**Obesity; n** (%)	85 (47.2%)	30 (28.3%)	55 (74.3%)
**GH therapy**			
*Never*	30 (16.7%)	9 (8.5%)	21 (28.4%)
*Current or past*	150 (83.3%)	97 (91.5%)	53 (71.6%)
**Age at start of GH therapy; yrs**	2.5 [0.9; 6.5] (0.2 to 32)	1.3 [0.7; 2.7] (0.2 to 11.7)	8.5 [3.9; 12] (0.6 to 32)
**Duration of GH therapy; yrs**	6.3 [2.8; 9.8] (0.1 to 20.3)	5.3 [2.4; 8.5] (0.1 to 14.2)	9 [3.3; 11.9] (0.5 to 20.3)

Data are expressed as number (%), mean ± standard deviation (range) or median (25th; 75th percentile) (range); yrs: years; PWS: Prader–Willi syndrome; del15: paternal deletion of chromosome 15; UPD15: maternal uniparental disomy for chromosome 15; BMI: Body mass index; SDS: standard deviation scores; GH: growth hormone.

**Table 2 jcm-11-01574-t002:** Analysis of potential predictive factors for scoliosis in PWS patients.

	Without Scoliosis	With Scoliosis	HR [95% C.I.]	*p*-Value
**Number**	30 (17%)	150 (83%)		
**Genotype**				
*del15*	14 (46.7%)	85 (56.7%)	ref	-
*UPD15*	14 (46.7%)	64 (42.7%)	1.1 [0.79 to 1.53]	0.561
*Other*	2 (6.7%)	1 (0.7%)	0.3 [0.04 to 2.17]	0.234
**Gender**				
*Male*	16 (53.3%)	68 (45.3%)	ref	-
*Female*	14 (46.7%)	82 (54.7%)	1.29 [0.93 to 1.78]	0.127
**Therapy with GH**				
*Never treated or started therapy after scoliosis diagnosis*	5 (16.7%)	64 (42.7%)	ref	-
*Yes (before scoliosis diagnosis)*	25 (83.3%)	86 (57.3%)	1.08 [0.78 to 1.78]	0.655

Data are expressed as number (%); del15: paternal deletion of chromosome 15; UPD15: maternal uniparental disomy for chromosome 15; GH: growth hormone; PWS: Prader–Willi syndrome; HR: Hazard Ratio; C.I. Confidence Interval.

**Table 3 jcm-11-01574-t003:** Clinical and anthropometric measurements of PWS patients at final evaluation according to scoliosis.

	Without Scoliosis	With Scoliosis	*p*-Value
**Number**	30 (16.7%)	150 (83.3%)	
**Age at observation; yrs**	10.8 [3.5; 23.3] (1.7 to 41.4)	16.5 [9; 25.7] (1.7 to 49.7)	0.022
**Pubertal stage**			0.088
*1*	16 (53.3)	45 (30)	
*2*	2 (6.7)	25 (16.7)	
*3*	7 (23.3)	38 (25.3)	
*4*	5 (16.7)	42 (28)	
**BMI** (kg/m^2^)	26.8 ± 12.9 (13.8 to 64.6)	29.5 ± 10.9 (12.5 to 59.7)	0.302
**BMI SDS**	1.4 ± 1.8 (−1.9 to 4.5)	1.7 ± 1.6 (−3.8 to 4.4)	0.310

Data are expressed as number (%), mean ± standard deviation (range) or median (25th; 75th percentile) (range); yrs: years; BMI: Body mass index; SDS: standard deviation scores. PWS: Prader–Willi syndrome.

**Table 4 jcm-11-01574-t004:** Characteristics of 150 PWS patients stratified by severity of scoliosis.

	Mild	Moderate	Severe	Overall *p*-Value
**Number**	58 (38.7%)	43 (28.7%)	49 (32.6%)	
**Genetics**				1
*del15*	33 (56.9%)	24 (55.8%)	28 (57.1%)	
*UPD15*	24 (41.4%)	19 (44.2%)	21 (42.9%)	
*Other*	1 (1.7%)	0 (0%)	0 (0%)	
**Gender**				0.025
*Male*	33 (56.9%)	20 (46.5%)	15 (30.6%)	
*Female*	25 (43.1%)	23 (53.5%)	34 (69.4%)	
**Therapy with GH**				0.302
*Never treated or started therapy after scoliosis diagnosis*	24 (41.4%)	15 (34.9%)	25 (51%)	
*Yes (before scoliosis diagnosis)*	34 (58.6%)	28 (65.1%)	24 (49%)	
**Age at start of GH therapy; yrs**	2.3 [0.8; 6.6] (0.3 to 32)	2.1 [0.8; 7.8] (0.2 to 21)	2.7 [1.2; 6.8] (0.5 to 24)	0.830
**Age at diagnosis of scoliosis; yrs**	6.2 [3; 12.6] (0.8 to 43.4)	6.5 [4.7; 12.4] (0.9 to 30.4)	3.8 [2.1; 6.1] (0.7 to 22.2)	0.005
**BMI** (kg/m^2^)	28.9 ± 11.1 (12.6 to 54.9)	29 ± 10.8 (12.5 to 57.7)	30.6 ± 10.9 (13.5 to 59.7)	0.695
**BMI SDS**	1.69 ± 1.7 (−3.84 to 4.42)	1.58 ± 1.51 (−3.01 to 3.96)	1.84 ± 1.43 (−2.02 to 4.01)	0.726
**Obesity; n (%)**	30 (51.7%)	17 (39.5%)	26 (53.1%)	0.362

Data are expressed as number (%), mean ± standard deviation (range) or median (25th; 75th percentile) (range); del15: paternal deletion of chromosome 15; UPD15: maternal uniparental disomy for chromosome 15; GH: growth hormone; yrs: years; BMI: Body mass index; SDS: standard deviation scores; PWS: Prader–Willi syndrome.

**Table 5 jcm-11-01574-t005:** Clinical characteristics and anthropometric measurements of 26 PWS patients who underwent surgery.

Variables	Sample Characteristics
**Age at surgery** (yrs)	13.1 [10.6; 14.8] (5 to 27.5)
**Gender**	
*Male*	9 (34.6%)
*Female*	17 (65.4%)
**Genotype**	
*del15*	15 (57.7%)
*UPD15*	11 (42.3%)
**GH treatment (before surgery)**	
*No*	2 (7.7%)
*Yes*	24 (92.3%)
**Age at start of GH therapy; yrs**	2.7 [1.8; 8.5] (0.5 to 16)
**Duration of GH therapy; yrs**	9.3 ± 4.05 (1.3 to 16.8)
**Age at diagnosis of scoliosis; yrs**	3.4 [2.3; 6] (0.9 to 14.5)
**Brace** *(before surgery)*	25 (96.2%)
**BMI** (kg/m^2^)	29.3 ± 8.4 (17.3 to 52.5)
**BMI SDS**	1.75 ± 1.22 (−1.22 to 3.97)
**Obesity; n (%)**	13 (50%)

Data are expressed as number (%), mean ± standard deviation (range) or median (25th; 75th percentile) (range); yrs: years; del15: paternal deletion of chromosome 15; UPD15: maternal uniparental disomy for chromosome 15; GH: growth hormone; BMI: Body mass index; SDS: standard deviation scores; PWS: Prader–Willi syndrome.

## Data Availability

The datasets generated during and/or analysed during the current study are not publicly available but are available from the corresponding author on reasonable request.

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
