# Peer review of "High Prevalence of Scoliosis in a Large Cohort of Patients with Prader-Willi Syndrome"

_jcm, 2022, doi:10.3390/jcm11061574_

Round 1

Reviewer 1 Report

Manuscript ID

jcm-1612225

Title

HIGH PREVALENCE OF SCOLIOSIS IN A LARGE COHORT OF PATIENTS WITH PRADER-WILLI SYNDROME

This article is about scoliosis in Prader –Willi syndrome. The paper aboards some questions about the real prevalence, gender predominance, relationship with genetics, GH treatment, age at diagnosis and prognosis of scoliosis linked to PWS. It’s well designed and answers some issues not yet solved in previous literature. I think it deserves to be published.

I have some comments:

-Introduction page 2, line 2: I am not sure if central adrenal insufficiency should be written here. It was classically proposed as a typical dysfunction in PWS, but different groups failed in demonstrating that.

-Page 2 line: you mention osteoporosis as a commonly manifestation in PWS. Could you specify if it’ls linked to hypogonadism? Please comment.

-Page 2. Study population. Please add code number for ethical committee acceptation. Also when you speak about consent form “when appropriate”: what do you mean? Perhaps if you have a Registry of this patients you asked for permission in the past?

-Table 1: You define 36% of children as Tanner, coud it correspond to adipomasty? It surprised me. Also, 55% of adult group ( adipomasty, under substitutive therapy treatment?) Please, explain.

-Figure 1. Kaplan-Meier. I understand about five years of live you have detected 50% of cases. Could you explain the first age scoliosis evaluation of the group? It’s unclear if this age could be lower in case you begun to study them before 2 years of age. Please clarify.

-Discussion: last parghraph. I am not sure if reference 2 is appropriately linked here.

Congratulations. The paper was very interesting to me.

Author Response

Responses to reviewer #1

Manuscript ID: jcm-1612225

Thank you for reviewing our manuscript and for your positive comments. We wish to express our appreciation for your insightful comments on our paper. We hope that the revised version of the manuscript can fully satisfy your requests. Text changes are highlighted in bold.

Q1. Introduction page 2, line 2: I am not sure if central adrenal insufficiency should be written here. It was classically proposed as a typical dysfunction in PWS, but different groups failed in demonstrating that.

A1 We completely agree with your comment. The text has been changed accordingly: (hypogonadism, growth hormone/insulin-like growth factor-I axis dysfunction, hypothyroidism, hyperghrelinemia and, rarely, central adrenal insufficiency)

Q2. Page 2 line: you mention osteoporosis as a commonly manifestation in PWS. Could you specify if it is linked to hypogonadism? Please comment.

A2 Thank you for your suggestion. The following sentence has been added in the text: “Incomplete pubertal development and low sex hormone levels are believed to be major causes of osteoporosis

Q3. Page 2. Study population. Please add code number for ethical committee acceptation.

A3. Thank you for your request. Code numbers for ethical committee acceptance have been added:  “Both the Ethical Committee of Bambino Gesù Children’s Hospital, Rome and the Ethical Committee of Istituto Auxologico Italiano, IRCCS, Milan, Italy approved the study protocol (research code: PWS_ISS, acronym: 2092/2020, and research code: 2014_01_21_06; acronym: SCOLADUPWS, respectively).

Q4. Also when you speak about consent form “when appropriate”: what do you mean? Perhaps if you have a Registry of this patients you asked for permission in the past?

A4. Thank you for your comment. Our sentence is unclear and has been rephrased as follows: “Written informed consent was obtained from all parents or legal guardians, and from the patients when applicable”.

Q5. Table 1: You define 36% of children as Tanner, could it correspond to adipomasty? It surprised me. Also, 55% of adult group (adipomasty, under substitutive therapy treatment?) Please, explain.

A5. Thank you for your valuable comment. Our data are conditioned by the treatment of hypogonadism. According to the literature (Pellikaan et al., J Clin Med. 2021 Sep 24;10(19):4361; Pellikaan et al., J Clin Med. 2021 Dec 10;10(24):5781) a significant percentage of our patients underwent sex steroid therapy. The following sentence has been added in the paragraph “3.1. Characteristics of the cohort”: “Altogether, 53 patients (29.4%) underwent treatment with sex steroids”.

Q6. Figure 1. Kaplan-Meier. I understand about five years of live you have detected 50% of cases. Could you explain the first age scoliosis evaluation of the group? It’s unclear if this age could be lower in case you begun to study them before 2 years of age. Please clarify.

A6. Thank you for the opportunity to clarify this important aspect. In our retrospective cohort we found that median age at diagnosis of scoliosis was 6.3 years. This means that, in our experience, when reaching this age the cumulative probability of receiving a diagnosis of scoliosis raises to 0.5. Actually, as scoliosis is not an acute condition, this result is likely influenced by the timing of scoliosis assessment and the risk of having obtained a delayed estimate is thus possible. Unfortunately, due to the retrospective design of our study we do not have available all the assessments before diagnosis and we cannot evaluate the impact of the first age at scoliosis evaluation on our findings.

Q7. Discussion: last paragraph. I am not sure if reference 2 is appropriately linked here.

A7. You are right. The correct reference is [4].

Reviewer 2 Report

This retrospective cross-sectional study examining the prevalence of scoliosis in PWS syndrome is well designed as for examined parameters and considers a large cohort of patients. It strengthens the importance of orthopaedic follow-up throughout the lifespan. 

Just as an advice, tables are very informative but also a bit "full", maybe underlying in bold the titles inside it or better separating various parameters could help the reader to catch the information

Author Response

Responses to reviewer #2

Manuscript ID: jcm-1612225

Thank you for taking the time to review our article and for the appreciation of our work.

Q1 Just as an advice, tables are very informative but also a bit "full", maybe underlying in bold the titles inside it or better separating various parameters could help the reader to catch the information.

A1. Thank you for your suggestion. The Tables have been modified to make them more readable and clear.
